# Ceiling effect of clomiphene citrate on the testosterone to estradiol ratio in eugonadal infertile men

**Yian Liao, Yi-Kai Chang, Shuo-Meng Wang, Hong-Chiang Chang** *

Department of Urology, National Taiwan University Hospital, Taipei, Taiwan

* changhong@ntu.edu.tw

## Abstract

### Introduction

The testosterone to estradiol ratio (T/E2 ratio) reportedly exerts a stronger effect on semen quality and sexual desire than does testosterone alone. Clomiphene citrate is a selective estrogen receptor modulator that has long been used as an empirical treatment option in the management of idiopathic oligozoospermia. Clomiphene may change the hypothalamus–pituitary–gonad axis and result in the alteration of the T/E2 ratio. No reliable data are available regarding the change in the T/E2 ratio after clomiphene use in eugonadism.

### Methods

This study included 24 male patients who were diagnosed with idiopathic infertility with eugonadism. They all received clomiphene citrate (25 mg/day) as empirical treatment. Blood tests for serum testosterone, estradiol, prolactin, luteinizing hormone, and follicle stimulating hormone were performed before and after 4 weeks of clomiphene use. Paired t-tests were used to evaluate the significance of the hormone level change.

### Results

Overall, the patients' T/E2 ratio did not increase significantly after clomiphene use. In the subgroup analysis, the T/E2 ratio of patients with a baseline ratio of <200 increased significantly after clomiphene use.

### Conclusions

Clomiphene citrate may significantly increase the T/E2 ratio in eugonadal men under the premise of its ceiling effect (T/E2 ratio < 200), providing practitioners with guidance on the use of clomiphene in this demographic.

**Data Availability Statement:** All relevant data are within the manuscript.

**Funding:** The authors received no specific funding for this work.

**Competing interests:** The authors have declared that no competing interests exist.

**Abbreviations:** T/E2 ratio, Testosterone to Estradiol ratio; SERMs, Selective estrogen receptor modulators; GnRH, Gonadotropin-releasing hormone; FSH, Follicle-stimulating hormone; LH, Luteinizing hormone.

## Introduction

Selective estrogen receptor modulators (SERMs) are drugs that act on estrogen receptors [1]. Such drugs can be divided into receptor agonists and antagonists. These drugs selectively act on specific organs, with effects differing by organs [1]. Clomiphene citrate is both a nonsteroidal antiestrogen drug and an SERM that competes with estradiol for estrogen receptors in the hypothalamus and blocks the normal negative feedback of circulating estradiol on the hypothalamus [2]. The production of gonadotroptin-releasing hormone (GnRH) may not be limited by estrogen; instead, the pituitary gland releases more follicle-stimulating hormone (FSH) and luteinizing hormone (LH) and causes an increase in sperm and testosterone production by the testes [2]. High levels of intratesticular testosterone is a key factor in the nonconventional pathway of testosterone for spermatogenesis [3].

Abhyankar [4] revealed that the testosterone/estradiol (T/E2) ratio exerts a stronger effect on semen quality and sexual desire than did testosterone alone; the author concluded that a larger increase in the sperm concentration and total motility count was correlated with a larger increase in the posttreatment T/E2 ratio. A small population study by Shabsigh [5] also reported an increased T/E2 ratio after clomiphene use in patients with hypogonadism; the author concluded that low dose clomiphene citrate may improve the T/E2 ratio in men with hypogonadism. However, to our knowledge, no large study has been conducted on the T/E2 ratio change after clomiphene use in infertile patients with eugonadism. The aim of this preliminary study was to determine whether clomiphene citrate exerts an effect on the T/E2 ratio in infertile patients with eugonadism.

## Materials and methods

This single-center, retrospective study was conducted in the Urology Department of #### from May 1, 2018, to May 31, 2019. Inclusion criteria were as follows: (1) being a man diagnosed with infertility; (2) taking clomiphene as testosterone restoration therapy; (3) accepting blood tests including those for prolactin, LH, FSH, testosterone, and estrogen; and (4) having an initial testosterone level within the normal range. Those who had undergone surgery of the testis or had received hormonal drugs that would affect the testosterone level (including dutasteride and leuprorelin) were excluded. Infertility was defined as failure to establish a clinical pregnancy after 12 months of regular and unprotected sexual intercourse. The normal range of the total testosterone level was defined as 240 to 870 ng/dL.

The study protocol was approved by the Institutional Review Board of #### with relevant jurisdiction (IRB No. 201903111RIN). The necessity of informed consent was confirmed by the institutional review board.

Before receiving clomiphene citrate treatment, blood samples were examined to record baseline serum total testosterone, estradiol, LH, FSH, and prolactin levels. All patients received clomiphene citrate 25 mg once a day for at least 1 month. Patients were followed up at 4 weeks after the treatment initiation to evaluate the treatment response. At the follow-up visit, the aforementioned serum levels were recorded to compare with those before clomiphene use. All hormone test(testosterone, estradiol, LH, FSH and prolactin) included in our study are sent to the central lab in "Department of Laboratory Medicine of National Taiwan University Hospital". FSH, LH, E2 and prolactin were measured by using IMMULITE® 2000 immunoassay system and Testosterone was measured by the ARCHITECT® 2nd Generation Testosterone assay which is a chemiluminescent microparticle immunoassay for the quantitative determination of testosterone. Simple statistical analysis (paired t-tests) was used to analyze hormone level changes resulting from clomiphene therapy. The primary outcome of our study was changes in the T/E2 ratio after the administration of clomiphene citrate in eugonadal men

with infertility. Secondary outcomes included changes in other hormone levels (FSH, LH, and prolactin).

## Results

In total, 22 men were retrospectively reviewed in this study. The average age of these patients was 40.9 (range: 31–48) years, with 12 being older than 40 years. All patients were followed up in an outpatient clinic by one urologist. Most of these patients were healthy individuals seeking treatment for infertility; however, one had hypertension and ulcerative colitis, and another had liver cirrhosis, diabetes mellitus, and polyneuropathy.

The mean total testosterone level before treatment (n = 22) was 469 ± 204 (range: 212–947) ng/dL, and the mean estradiol level (n = 21) was 3.819 ± 1.312 (range: 1.3–7.8) ng/dL. The mean T/E2 ratio at the first visit (n = 21) was 160 ± 90 (Table 1). The mean LH (n = 21) and FSH (n = 22) levels were 4.89 ± 3.05 (range: 1.54–14.3) mIU/mL and 5.68 ± 3.17 (range: 1.59–14.4) mIU/mL, respectively. The mean prolactin level (n = 21) was 6.80 ± 3.32 (range: 0.9–12.7) ng/mL (Table 1).

The patients were re-evaluated approximately 4 weeks after initiating therapy with 25 mg of oral clomiphene citrate once a day. A significant increase was noted in the mean total testosterone level (n = 22) to 935 ± 349 (range: 285–1916) ng/dL, representing an increase of 99.2% (P < 0.001). The mean estradiol level (n = 21) increased significantly to 6.605 ± 3.091 (range: 2.57–1.30) ng/dL, representing a rise of 80.8% (P < 0.001). The T/E2 ratio also increased from 160 ± 90 to 180 ± 60, a rise of 12.5%; however, the difference was not significant (P = 0.302; Table 1). The posttreatment mean LH (n = 21) and FSH (n = 22) levels increased significantly to 10.90 ± 5.92 (range: 2.93–22.2) mIU/mL (P < 0.001) and 11.43 ± 7.86 (range: 4.25–36.9) mIU/mL (P = 0.002), representing an increase of 123.1% and 101.3%, respectively. The posttreatment mean prolactin level (n = 21) decreased to 5.77 ± 2.49 (range: 2.84–12.20) ng/mL; this 15.1% decrease, however, did not reach significance (P = 0.297; Table 1).

The 21 patients included were further analyzed according to the baseline T/E2 ratio. Four patients had a baseline T/E2 ratio of <100. Their mean T/E2 ratio increased significantly from 070 ± 10 to 110 ± 20 after clomiphene use, representing an increase of 57.1% (P = 0.006). Twelve patients had a baseline T/E2 ratio of < 150. Their mean T/E2 ratio significantly increased from 090 ± 30 to 130 ± 40 after clomiphene use, which is an increase of 44.4% (P = 0.006). Fifteen patients had a baseline T/E2 ratio of <200. Their mean T/E2 ratio significantly increased from 120 ± 40 to 150 ± 50 after clomiphene use, representing an increase of 25.0% (P = 0.0017). The T/E2 ratio of the five patients with a baseline T/E2 ratio < 250

**Table 1. Endocrinology data at baseline and 4 weeks after initiating clomiphene treatment.**

| Hormone (Conc.) | Number of cases | Baseline[a] | Post treatment[b] | Clomiphene effect | P value |
|---|---|---|---|---|---|
| Testosterone(ng/dL) | 22 | 469±204 | 935±349* | ↑99.2% | <0.001 |
| Estradiol (ng/dL) | 21 | 3.819±1.312 | 6.605±3.091* | ↑80.8% | <0.001 |
| T/E2 ratio | 21 | 160±90 | 180±60 | ↑ 12.5% | 0.302 |
| LH (mIU/mL) | 21 | 4.89±3.05 | 10.90±5.92* | ↑ 123.1% | <0.001 |
| FSH (mIU/mL) | 22 | 5.68±3.17 | 11.43±7.86* | ↑101.3% | 0.002 |
| Prolactin (ng/mL) | 21 | 6.80±3.32 | 5.77±2.49 | ↓15.1% | 0.297 |

Values are presented as the mean±standard deviation.

*Versus baseline data (p<0.05).

[a]Baseline hormone level before Clomiphene use.

[b]Hormone level after Clomiphene use.

increased from 120 ± 50 to 160 ± 50; however, the difference was not significant (P = 0.09), which is similar to the result in the group with all patients included (n = 21, P = 0.302; Table 2).

## Discussion

Clomiphene citrate, an SERM, is a nonsteroidal antiestrogen drug that has empirically been used in the management of idiopathic oligospermia. The drug competes with estradiol for estrogen receptors in the hypothalamus and blocks the normal negative feedback of circulating estradiol on the hypothalamus [6, 7]. Under clomiphene therapy, the amplitude of GnRH pulses increases, stimulating the pituitary gland to produce more FSH and LH. As a result, testicular total testosterone also increases [6, 8–10]. When used to treat male infertility, clomiphene is well tolerated with no identified serious adverse effects [11–13].

Few studies have investigated the relationship between clomiphene citrate use and the T/E ratio. Shabsigh [4] recruited 36 Caucasian men with hypogonadism, defined as having a serum total testosterone level of <300 ng/dL. Each patient was treated with a daily dose of 25 mg clomiphene citrate. The serum levels of total testosterone and estradiol were recorded at baseline and follow-up visits. By the first follow-up visit (at 4–6 weeks), the mean total testosterone level had increased significantly (P < 0.00001). Moreover, the T/E ratio improved from 8.7 to 14.2 (P < 0.001). Thus, the T/E2 ratio of patients with hypogonadism increased after clomiphene use. No further research on clomiphene use and the testosterone to estradiaol ratio in eugonadal patients has been conducted, prompting the current study.

In our study, we observed a significant increase in the mean total testosterone level, with an increase of 99.2% (P < 0.001). The mean estradiol level also increased significantly, with a rise of 80.8% (P < 0.001). These results are similar to those of patients with hypogonadism in the aforementioned study [4] who took clomiphene citrate. As for other hormone levels, the mean FSH and LH levels increased significantly after clomiphene treatment, with a rise of 101.3% (P = 0.002) and 123.1% (P < 0.001), respectively. Because clomiphene citrate competes with estradiol for estrogen receptors in the hypothalamus, it blocks the normal negative feedback of circulating estradiol in the hypothalamus. As a result, LH and FSH levels would increase, which is consistent with our results.

Regarding the T/E2 ratio, compared with the T/E2 ratio in the general male population, that of patients enrolled in our study was relatively low. Gong [14] enrolled 337 patients in a single center as the control group to study the correlation between the T/E2 ratio and cardiovascular events, revealing the normal range of the T/E2 ratio to be 190 ± 60. In our study, the

**Table 2. T/E2 ratio of men with a lower baseline ratio (<0.20) increased significantly after clomiphene treatment.**

| Upper limit of T/E2 | Number of cases | Baselineª | Post treatmentᵇ | Clomiphene effect | P value |
|---|---|---|---|---|---|
| Any | 21 | 160±90 | 0.18±0.06 | ↑12.5% | 0.302 |
| <0.25 | 17 | 120±50 | 160±50 | ↑33.3% | 0.09 |
| <0.20 | 15 | 120±40 | 150±50* | ↑25.0% | 0.017 |
| <0.15 | 12 | 90±30 | 130±40* | ↑44.4% | 0.006 |
| <0.10 | 4 | 70±10 | 110±20* | ↑57.1% | 0.006 |

Values are presented as the mean±standard deviation.

*Versus baseline data (p<0.05).

ª Baseline T/E2 level before clomiphene use.

ᵇ T/E2 after clomiphene use.

mean baseline T/E2 ratio was 160 ± 90. When including all patients, we noted an increase of 6.25% in the ratio, but the difference was not significant. We obtained different results when setting an upper limit of the T/E2 ratio. The mean T/E2 ratio increased significantly (57.1%) after clomiphene use if only patients with a T/E2 ratio of <100 were included. A significant increase in the T/E2 ratio after clomiphene use was also noted in those with a T/E2 ratio of <200. After slightly increasing the upper limit to a T/E2 ratio of <250, we still noted an increase of 33.3% in the T/E2 ratio; however, the difference was not significant. Therefore, we suggest ceiling effects for increasing the T/E2 ratio; in the current study, the ceiling effect was set at a T/E2 ratio of <200. This result suggests that using clomiphene in infertile men with eugonadism whose T/E ratio is >200 is not advisable.

This study has some limitations. Only 24 patients were included; thus, the sample was relatively small. Moreover, because this was a retrospective study, the exposure method could not be controlled, potentially resulting in variable clomiphene citrate dosage and frequency. Possible factors that may affect total testosterone and estradiol level, such as BMI level, waistline and testicular volume, were also not included. Future studies should prospectively include these parameters for analysis to confirm the T/E2 ratio changeafter clomiphene citrate use in infertile men with eugonadism.

## Conclusion

Clomiphene citrate may significantly increase the T/E2 ratio in eugonadal infertile men under the premise of its ceiling effect (T/E2 ratio < 200), providing useful guidance for the use of clomiphene in this group of patients. Further investigations, especially those with a larger sample size, are required to confirm the results.

## Acknowledgments

The authors also wish to thank all patients participated in this study.

## Author Contributions

**Conceptualization:** Yian Liao, Shuo-Meng Wang, Hong-Chiang Chang.

**Data curation:** Yian Liao.

**Formal analysis:** Yian Liao.

**Investigation:** Yian Liao.

**Methodology:** Yian Liao.

**Resources:** Yian Liao.

**Software:** Yian Liao.

**Supervision:** Yi-Kai Chang, Shuo-Meng Wang, Hong-Chiang Chang.

**Writing – original draft:** Yian Liao.

**Writing – review & editing:** Yian Liao, Hong-Chiang Chang.

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
