## [Decision Letter · Decision Letter 0]

17 Nov 2021

PONE-D-21-24474Ceiling Effect of Clomiphene Citrate on the Testosterone to Estradiol Ratio in Eugonadal Infertile MenPLOS ONE

Dear Dr. HC Chang,

Thank you for submitting your manuscript to PLOS ONE. After careful consideration, we feel that it has merit but does not fully meet PLOS ONE’s publication criteria as it currently stands. Therefore, we invite you to submit a revised version of the manuscript that addresses the points raised during the review process.

Please submit your revised manuscript at your earliest convenience. Please include the following items when submitting your revised manuscript:A rebuttal letter that responds to each point raised by the academic editor and reviewer(s). You should upload this letter as a separate file labeled 'Response to Reviewers'.A marked-up copy of your manuscript that highlights changes made to the original version. You should upload this as a separate file labeled 'Revised Manuscript with Track Changes'.An unmarked version of your revised paper without tracked changes. You should upload this as a separate file labeled 'Manuscript'.

We look forward to receiving your revised manuscript.

Kind regards,

Joël R Drevet, Ph.D.

Academic Editor

PLOS ONE

Journal Requirements:

Additional Editor Comments (if provided):

Your submission has been evaluated by two independent reviewers who found that it has potential but requires substantial revision. Please revise your manuscript by addressing, to the best of your ability, each of the reviewers' concerns and providing additional material to strengthen your report.

Reviewers' comments:

Reviewer's Responses to Questions

**Comments to the Author**

1. Is the manuscript technically sound, and do the data support the conclusions?

Reviewer #1: No

Reviewer #2: Yes

2. Has the statistical analysis been performed appropriately and rigorously? 

Reviewer #1: I Don't Know

Reviewer #2: Yes

3. Have the authors made all data underlying the findings in their manuscript fully available?

Reviewer #1: Yes

Reviewer #2: No

4. Is the manuscript presented in an intelligible fashion and written in standard English?

Reviewer #1: Yes

Reviewer #2: Yes

5. Review Comments to the Author

Reviewer #1: I appreciate the manuscript and the work that went into preparing it. Just a few suggestions to strengthen the product:

1. As aromatase breaks down T to E2, would report on the effect of BMI on results/outcomes.

2. Should the Cirrhosis patient be excluded since liver disease effects sex hormone binding globulin production (effects total Testosterone)?

3. Is the standard to use ng/mL for T? I'm more used to ng/dL...with the normal T:E ratio being 10:1.

Best regards.

Reviewer #2: Dear Dr. Hong-Chiang Chang

Based on my own reading of your paper, I have some issues I want to address. I would like to say right away that the theme “ceiling effect” is quite interesting as a predictor for clomiphene citrate (CC) use.

I emphasize that the PDF of the article I received does not have line:numbers, which made the review a bit difficult.

1) Introduction: I did not observe references for all the information contained in the paragraph below. Despite being general domain information, it is important to always add reference sources, even if it is in the introduction.

“Selective estrogen receptor modulators (SERMs) are drugs that act on estrogen receptors. Such drugs can be divided into receptor agonists and antagonists (Ref?). These drugs selectively act on specific organs, with effects differing by organs (Ref?). Clomiphene citrate is both a nonsteroidal antiestrogen drug and an SERM that competes with estradiol for estrogen receptors in the hypothalamus and blocks the normal negative feedback of circulating estradiol on the hypothalamus (Ref?). The production of gonadotroptin-releasing hormone (GnRH) may not be limited by estrogen; instead, the pituitary gland releases more follicle-stimulating hormone (FSH) and luteinizing hormone (LH) and causes an increase in sperm and testosterone production by the testes (Ref?). High levels of intratesticular testosterone is a key factor in the nonconventional pathway of testosterone for spermatogenesis (Ref?).

2) Materials and Methods

I would like the authors to specify which reference was used to define the diagnosis of eugonadism (total testosterone 2.4 to 8.7 ng/mL).

The laboratory diagnosis of hypogonadism is based on the measurement of total testosterone. The American Society of Endocrinology recommends the value of 264 ng/dL (9.2 nmol/L) as the lower limit to define the hypogonadal state; however, the cutoff levels for diagnosis are not a consensus. So I would like the authors to specify the Guideline used.

(Testosterone Therapy in Men With Hypogonadism: An Endocrine Society* Clinical Practice Guideline)

I would like the authors to write “range of the total testosterone level” instead of just “range of the testosterone level” – here it can be confusing if total or free testosterone was used as a parameter.

I would like more detail in the table of 22 patients at baseline with data about BMI, waistline and testicular volume if possible

3) Discussion

An important point to be considered is the ability of the CC to induce a response in patients with central or mixed hypogonadism. In the different studies in which its use was evaluated, it was possible to define response (elevation of testosterone) in patients with Leydig cells capable of reacting to the central stimulus restored by CC (that's why I would like to know about testicular volume).

It is known that the unbalance of the T:E2 ratio is directly implicated in the pathophysiology of hypogonadism. It is important to emphasize that the increase in estrogen with the use of CC results from the aromatization of testosterone in the adipose tissue, especially the visceral tissue. I would like to see the details of the abdominal fat of these patients in order to understand whether the ceiling effect proposed by the authors is related to patients with larger abdominal fat/waistline and therefore greater visceral fat.

6. PLOS authors have the option to publish the peer review history of their article (what does this mean?). If published, this will include your full peer review and any attached files.

Reviewer #1: No

Reviewer #2: **Yes: **Andressa Heimbecher Soares

---

## [Author Response · Author response to Decision Letter 0]

30 Nov 2021

*To editor:

We really appreciate for reminding us all additional journal requirements. The article has now met PLOS ONE's style requirements. As for the Data Availability statement, we stated that there are no supporting information in the main body of our article.

*To reviewer 1: 

Q: 1. As aromatase breaks down T to E2, would report on the effect of BMI on results/outcomes.

Answer: Different BMI levels may probably affect T/E2 ratio. However, we did not include patients’ BMI in our data base. Your suggestion is good for further study of this topic and data of BMI would make a more precise analysis.

Q: 2. Should the Cirrhosis patient be excluded since liver disease effects sex hormone binding globulin production (effects total Testosterone)?

Answer: Cirrhosis surely affect sex hormone binding globulin production, indirectly influencing the bioavailability of testosterone. As for the patient included in our study with liver cirrhosis, his Child-Pugh score is only grade A with normal bilirubin, PT/APTT and albumin level, which may hardly have an impact on sex hormone binding globulin production. 

Q: 3. Is the standard to use ng/mL for T? I'm more used to ng/dL...with the normal T:E ratio being 10:1.

Answer: There may not be a standard unit of testosterone level. According to EAU(European Association of Urology) guideline, ng/mL and nmol/L have both been used as the unit of testosterone level. 

*To reviewer 2:

1) Introduction: 

Q: I did not observe references for all the information contained in the paragraph below. Despite being general domain information, it is important to always add reference sources, even if it is in the introduction.

“Selective estrogen receptor modulators (SERMs) are drugs that act on estrogen receptors. Such drugs can be divided into receptor agonists and antagonists (Ref?). These drugs selectively act on specific organs, with effects differing by organs (Ref?). Clomiphene citrate is both a nonsteroidal antiestrogen drug and an SERM that competes with estradiol for estrogen receptors in the hypothalamus and blocks the normal negative feedback of circulating estradiol on the hypothalamus (Ref?). The production of gonadotroptin-releasing hormone (GnRH) may not be limited by estrogen; instead, the pituitary gland releases more follicle-stimulating hormone (FSH) and luteinizing hormone (LH) and causes an increase in sperm and testosterone production by the testes (Ref?). High levels of intratesticular testosterone is a key factor in the nonconventional pathway of testosterone for spermatogenesis (Ref?).

Answer: We really appreciate the advice of adding references. Three references have been added to the introduction. The following paragraph is the revised introduction.

“Selective estrogen receptor modulators (SERMs) are drugs that act on estrogen receptors and such drugs can be divided into receptor agonists and antagonists [1]. These drugs selectively act on specific organs, with effects differing by organs [1]. Clomiphene citrate is both a nonsteroidal antiestrogen drug and an SERM that competes with estradiol for estrogen receptors in the hypothalamus and blocks the normal negative feedback of circulating estradiol on the hypothalamus [2]. The production of gonadotroptin-releasing hormone (GnRH) may not be limited by estrogen; instead, the pituitary gland releases more follicle-stimulating hormone (FSH) and luteinizing hormone (LH) and causes an increase in sperm and testosterone production by the testes [2]. High levels of intratesticular testosterone is a key factor in the nonconventional pathway of testosterone for spermatogenesis [3].”

Reference:

1. Chua, M.E., et al. Revisiting oestrogen antagonists (clomiphene or tamoxifen) as medical empiric therapy for idiopathic male infertility: a meta-analysis. Andrology, 2013. 1: 749.

2. Ribeiro, R.S., et al. Clomiphene fails to revert hypogonadism in most male patients with conventionally treated nonfunctioning pituitary adenomas. Arq Bras Endocrinol Metabol, 2011. 55: 266.

3. Hussein, A., et al. Clomiphene administration for cases of nonobstructive azoospermia: a multicenter study. J Androl, 2005. 26: 787.

2) Materials and Methods

Q: I would like the authors to specify which reference was used to define the diagnosis of eugonadism (total testosterone 2.4 to 8.7 ng/mL).

The laboratory diagnosis of hypogonadism is based on the measurement of total testosterone. The American Society of Endocrinology recommends the value of 264 ng/dL (9.2 nmol/L) as the lower limit to define the hypogonadal state; however, the cutoff levels for diagnosis are not a consensus. So I would like the authors to specify the Guideline used.

(Testosterone Therapy in Men With Hypogonadism: An Endocrine Society* Clinical Practice Guideline)

Answer: Exactly I agree that the cutoff level of hypogonadism and eugonadism has not reached a consensus. Different hospital may have different lab reference level with regard to their laboratory department. All patient included in our study have done blood examination in National Taiwan University Hospital. The reference range of total testosterone(2.4 to 8.7 ng/mL) is defined by Department of Laboratory Medicine of National Taiwan University Hospital. 

Q: I would like the authors to write “range of the total testosterone level” instead of just “range of the testosterone level” – here it can be confusing if total or free testosterone was used as a parameter.

Answer: We really appreciate the advice of modifying “testosterone” to “total testosterone” not to be confusing. Adjustment have been made throughout the whole manuscript(changing the testosterone level to the total testosterone level)

Q: I would like more detail in the table of 22 patients at baseline with data about BMI, waistline and testicular volume if possible

Answer: BMI level, waistline and testicular volume are surely the possible factor that may affect total testosterone and estradiol level, indirectly influencing the T/E2 ratio. Unfortunately, these parameters were not included in our study. Due to the retrospective nature of our study, these factors were not recorded in our data base. Further prospective study including these factors may be needed. 

3) Discussion

Q: An important point to be considered is the ability of the CC to induce a response in patients with central or mixed hypogonadism. In the different studies in which its use was evaluated, it was possible to define response (elevation of testosterone) in patients with Leydig cells capable of reacting to the central stimulus restored by CC (that's why I would like to know about testicular volume).

It is known that the unbalance of the T:E2 ratio is directly implicated in the pathophysiology of hypogonadism. It is important to emphasize that the increase in estrogen with the use of CC results from the aromatization of testosterone in the adipose tissue, especially the visceral tissue. I would like to see the details of the abdominal fat of these patients in order to understand whether the ceiling effect proposed by the authors is related to patients with larger abdominal fat/waistline and therefore greater visceral fat.

Answer: According to EAU (European Association of Urology) guideline, larger abdominal fat and greater visceral fat has now been established to be related to hypogonadism. Increase in estrogen by aromatase in visceral fat has also been established in several studies. Unfortunately, these factors were also not included in our study and may not be achieved due to the retrospective nature of our study. We will work our best on the further prospective study of relevant topics.

---

## [Decision Letter · Decision Letter 1]

30 Dec 2021

PONE-D-21-24474R1Ceiling Effect of Clomiphene Citrate on the Testosterone to Estradiol Ratio in Eugonadal Infertile MenPLOS ONE

Dear Dr. Hong-Chiang Chang,

Thank you for submitting your manuscript to PLOS ONE. After careful consideration, we feel that it has merit but does not fully meet PLOS ONE’s publication criteria as it currently stands. Therefore, we invite you to submit a revised version of the manuscript that addresses the points raised during the review process.

We look forward to receiving your revised manuscript.

Kind regards,

Joël R Drevet, Ph.D.

Academic Editor

PLOS ONE

Additional Editor Comments:

Although your manuscript has been improved, there are still aspects that require the authors' attention.

Reviewers' comments:

Reviewer's Responses to Questions

**Comments to the Author**

1. If the authors have adequately addressed your comments raised in a previous round of review and you feel that this manuscript is now acceptable for publication, you may indicate that here to bypass the “Comments to the Author” section, enter your conflict of interest statement in the “Confidential to Editor” section, and submit your "Accept" recommendation.

Reviewer #2: All comments have been addressed

2. Is the manuscript technically sound, and do the data support the conclusions?

Reviewer #2: Yes

3. Has the statistical analysis been performed appropriately and rigorously? 

Reviewer #2: Yes

4. Have the authors made all data underlying the findings in their manuscript fully available?

Reviewer #2: Yes

5. Is the manuscript presented in an intelligible fashion and written in standard English?

Reviewer #2: Yes

6. Review Comments to the Author

Reviewer #2: Dear Dr Hong-Chiang Chang

Based on my own reading of your Manuscript Draft - Revision 1, I have some issues in particular I wish to address.

1) In this part of the manuscript, I would like to correct “A small population study by Shabsigh [5] also reported an increased T/E ratio after clomiphene use in patients with hypogonadism” to “A small population study by Shabsigh [5] also reported an increased T/E2 ratio after clomiphene use in patients with hypogonadism”. The number 2 on E was missing.

2) I would like the description of the laboratory kits used for the analysis of total testosterone level, estradiol level, LH, FSH and prolactin level to be detailed.

3) I would like to know if the dosage of sex hormone binding globulin (SHBG) levels was measured via morning lab draws.

4) To make it easy in terms of standardization, I would like to put the total testosterone level in ng/dL and estradiol levels in ng/dL (https://doi.org/10.1016/j.juro.2017.02.2652)

5) In the “limitations” part of the study, I would like it to be detailed that there wasn't possible to get BMI level, waistline and testicular volume data. I suggest that the authors should proceed with the study to evaluate obese men versus men with normal weight and analyse if there are differences in the ceiling effect.

6) If I may suggest, a greater number of participants would improve the strength of the results

7) The study generates an interesting hypothesis about which subgroup of patients may benefit most from the use of clomiphene citrate

7. PLOS authors have the option to publish the peer review history of their article (what does this mean?). If published, this will include your full peer review and any attached files.

Reviewer #2: **Yes: **Andressa Heimbecher Soares

---

## [Author Response · Author response to Decision Letter 1]

3 Jan 2022

Reviewer #2: 

Q: 1) In this part of the manuscript, I would like to correct “A small population study by Shabsigh [5] also reported an increased T/E ratio after clomiphene use in patients with hypogonadism” to “A small population study by Shabsigh [5] also reported an increased T/E2 ratio after clomiphene use in patients with hypogonadism”. The number 2 on E was missing.

A: We really appreciate for checking through the manuscript for any missing word or number. “Number 2” has now been added to T/E ratio. 

Q: 2) I would like the description of the laboratory kits used for the analysis of total testosterone level, estradiol level, LH, FSH and prolactin level to be detailed.

A: We add the following description about the measurement methods in my document. All hormone test(testosterone, estradiol, LH, FSH and prolactin) included in our study are sent to the central lab in “Department of Laboratory Medicine of National Taiwan University Hospital”. FSH, LH, E2 and prolactin were measured by using IMMULITE® 2000 immunoassay system and Testosterone was measured by the ARCHITECT® 2nd Generation Testosterone assay which is a chemiluminescent microparticle immunoassay for the quantitative determination of testosterone. 

Q:3) I would like to know if the dosage of sex hormone binding globulin (SHBG) levels was measured via morning lab draws.

A: Sex hormone binding globulin (SHBG) levels is surely a possible factor that may affect bioavailable testosterone, indirectly influencing the T/E2 ratio. Unfortunately, this parameter was not included in our study. Due to the retrospective nature of our study, this factor was not recorded in our data base. Further prospective study including this factor may be needed.

Q:4) To make it easy in terms of standardization, I would like to put the total testosterone level in ng/dL and estradiol levels in ng/dL (https://doi.org/10.1016/j.juro.2017.02.2652)

A: We really appreciate for this suggestion. All testosterone level and estradiol level have been changed to ng/dL. T/E2 ratio has also been adjusted due to the unit change of testosterone and estradiol. 

Q:5) In the “limitations” part of the study, I would like it to be detailed that there wasn't possible to get BMI level, waistline and testicular volume data. I suggest that the authors should proceed with the study to evaluate obese men versus men with normal weight and analyse if there are differences in the ceiling effect.

A: We appreciate for the advice of adding this part to out limitation. We surely will proceed with the study to include these parameters for analysis to see if there are differences in the ceiling effect.

Q:6) If I may suggest, a greater number of participants would improve the strength of the results

A: Thanks for reviewer’s nice suggestion. Small study population is surely a limitation of our study. Further studies will include more participants to strengthen our result. 

Q:7) The study generates an interesting hypothesis about which subgroup of patients may benefit most from the use of clomiphene citrate

A: We are glad that this study leads to an interesting result of “Clomiphene citrate significantly increase the T/E2 ratio in eugonadal infertile men under the premise of its ceiling effect (T/E2 ratio < 200)”. We would work hard on increasing the sample size and including more important parameters for further analysis.

---

## [Editor Report · Decision Letter 2]

10 Jan 2022

Ceiling Effect of Clomiphene Citrate on the Testosterone to Estradiol Ratio in Eugonadal Infertile Men

PONE-D-21-24474R2

Dear Dr. HC Chang,

We’re pleased to inform you that your manuscript has been judged scientifically suitable for publication and will be formally accepted for publication once it meets all outstanding technical requirements.

Kind regards,

Joël R Drevet, Ph.D.

Academic Editor

PLOS ONE
---

## [Editor Report · Acceptance letter]

21 Jan 2022

PONE-D-21-24474R2 

Ceiling Effect of Clomiphene Citrate on the Testosterone to Estradiol Ratio in Eugonadal Infertile Men 

Dear Dr. Chang:

I'm pleased to inform you that your manuscript has been deemed suitable for publication in PLOS ONE. Congratulations! Your manuscript is now with our production department. 

Kind regards, 

on behalf of

Prof. Joël R Drevet 

Academic Editor

PLOS ONE